# An ISO-certified genomics workflow for identification and surveillance of antimicrobial resistance

Norelle L. Sherry [1,2,3], Kristy A. Horan[1], Susan A. Ballard [1], Anders Gonçalves da Silva[1], Claire L. Gorrie [3], Mark B. Schultz [1], Kerrie Stevens[1], Mary Valcanis[1], Michelle L. Sait[1], Timothy P. Stinear [3], Benjamin P. Howden [1,2,3,4] ✉ & Torsten Seemann [1,3,4]

Realising the promise of genomics to revolutionise identification and surveillance of antimicrobial resistance (AMR) has been a long-standing challenge in clinical and public health microbiology. Here, we report the creation and validation of *abritAMR*, an ISO-certified bioinformatics platform for genomics-based bacterial AMR gene detection. The *abritAMR* platform utilises NCBI's *AMRFinderPlus*, as well as additional features that classify AMR determinants into antibiotic classes and provide customised reports. We validate *abritAMR* by comparing with PCR or reference genomes, representing 1500 different bacteria and 415 resistance alleles. In these analyses, *abritAMR* displays 99.9% accuracy, 97.9% sensitivity and 100% specificity. We also compared genomic predictions of phenotype for 864 *Salmonella* spp. against agar dilution results, showing 98.9% accuracy. The implementation of *abritAMR* in our institution has resulted in streamlined bioinformatics and reporting pathways, and has been readily updated and re-verified. The *abritAMR* tool and validation datasets are publicly available to assist laboratories everywhere harness the power of AMR genomics in professional practice.

Antimicrobial resistance (AMR) is an increasingly well-recognised threat to global health[1–3]. A clear understanding of the genomic and mechanistic basis for AMR is required to inform clinicians and public health teams, from the level of individual patients through to population-level surveillance[4,5]. By providing additional, timely data on acquired AMR genes or gene mutations that confer resistance, genomic sequencing has the potential to significantly enhance AMR surveillance and inform patient treatment beyond conventional phenotypic susceptibility testing methods[6,7].

The use of genomics in the detection and surveillance of bacterial AMR is lagging behind other applications of genomics, such as strain typing and phylogenetic analysis. Contributors to the lack of uptake include the fact that phenotypic testing can be performed more rapidly than genotypic testing for many common pathogens, and the correlation between genotype and phenotype can be variable due to incomplete knowledge of AMR mechanisms that impact function[4,8]. However, technological advances in whole genome sequencing (WGS) means the process is becoming more cost-effective and the turn-around time for sequencing a microbial genome is decreasing significantly[9].

A lack of international standards for genomic detection of AMR mechanisms means it is difficult to compare results between laboratories[10,11]. To facilitate implementation, the development of standardised and extensive open-access AMR databases and the

[1]Microbiological Diagnostic Unit Public Health Laboratory (MDU-PHL), Department of Microbiology & Immunology, University of Melbourne at the Peter Doherty Institute for Infection & Immunity, Melbourne, Victoria, Australia. [2]Department of Infectious Diseases, Austin Health, Heidelberg, Victoria, Australia. [3]Department of Microbiology & Immunology, University of Melbourne at the Peter Doherty Institute for Infection & Immunity, Melbourne, Australia. [4]These authors jointly supervised this work: Benjamin P. Howden, Torsten Seemann. ✉e-mail: bhowden@unimelb.edu.au

validation of bioinformatic analytical tools for the detection of AMR is crucial[4,12]. Another hurdle to the acceptance and implementation of AMR genomics is how the data can be meaningfully reported outside of research or reference laboratory settings[4]. If the implementation of WGS for AMR is going to be accepted for the detection of AMR resistance, it is important to consider the way in which complex genomic data is presented to clinicians, nurses, public health surveillance teams, and other stakeholders with varying understanding of genomics, and thus how to interpret findings[13]. This is a gap in the bioinformatic tools currently available for AMR, as outputs are not usually tailored for clinical reports, or easily modifiable to suit local reporting requirements.

In many countries, clinical and public health microbiology laboratories are required to meet International Standards Organization (ISO), or ISO-equivalent, standards to be accredited to operate[14]. These standards require the implementation of standardised operating procedures, quality management systems, staff training and rigorous validation of all processes used to generate results and reports in each laboratory[15]. Currently, the relevant standards for medical laboratories (last released in 2012) are not designed to assess the performance of bioinformatic tools, making validation to meet these standards difficult for laboratories, in addition to the paucity of publicly-available validation datasets[4].

Here we design and validate a bioinformatic tool, *abritAMR*, a wrapper for the NCBI *AMRFinderPlus* tool[16] for the detection of AMR determinants from whole genome sequencing data[16], with outputs adapted for clinical and public health microbiology reporting. We envisage that this pipeline, extensive validation methods and validation dataset could be adopted for use in public health and clinical sequencing laboratories to assist those involved in AMR surveillance and clinical applications.

## Results

### The *abritAMR* bioinformatics pipeline
*abritAMR* is a pipeline for characterisation and reporting of AMR determinants from bacterial sequences, adapting the *AMRFinderPlus* tool and database for use in clinical and public health microbiology. Using the outputs from *AMRFinderPlus*, AMR mechanisms are further classified by antimicrobial class and/or mechanism to suit clinical and public health microbiology (CPHM) needs, and subsequently filtered according to local reporting requirements, with results ready for incorporation into sample reports (overview Fig. 1, outputs Fig. 2; further details available in Methods and Supplementary Figure 1). An additional module generates inferred susceptibility results (currently validated for *Salmonella* spp.; output Fig. 2, reporting logic detailed in Supplementary Figure 2).

### Validation of the *abritAMR* pipeline: overview
To validate the *abritAMR* pipeline, we compared performance to PCR results for key AMR genes, to *AMRFinderPlus* results on synthetic read data from reference genomes, and to phenotypic data for *Salmonella* spp. (Fig. 3).

*abritAMR* performed very well against the four validation panels, with an overall accuracy of 99.9% (95% CI 99.9–99.9%), sensitivity 97.9% (97.5-98.4%), and specificity 100% (100-100%) (Table 1). Importantly, the *abritAMR* pipeline was reliable for the high-risk AMR gene

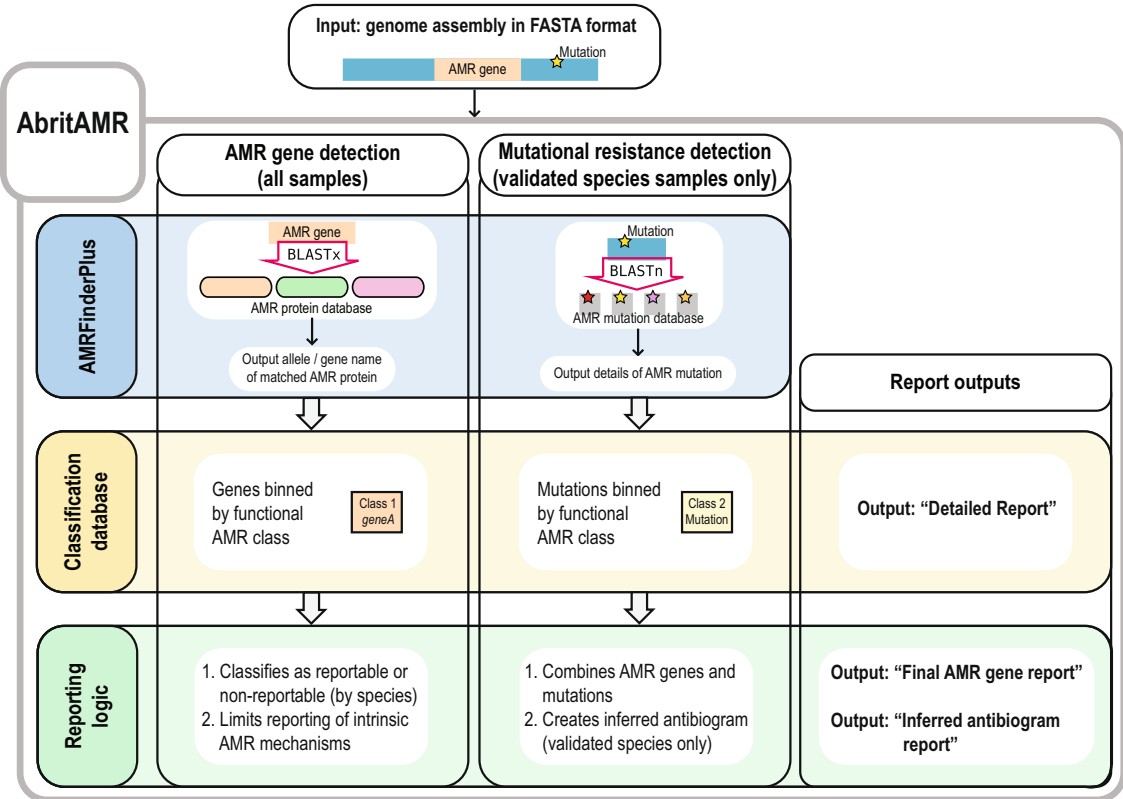

**Fig. 1 | Overview of *abritAMR* pipeline.** Assembled bacterial genomic sequence data (short- or long-read or hybrid assemblies, fasta format) are inputted in to the *abritAMR* pipeline to identify acquired AMR genes and mutations. The pipeline implements the *AMRFinderPlus* tool, identifying AMR genes (BLASTx search), and optionally identifies mutations associated with AMR for specified species where mutational data are available. Identified AMR genes +/- mutations are then binned into functional AMR classes according to the classification database (Detailed Report output, Fig. 2 and Supplementary Data 1). An additional step then applies reporting logic tailored to local requirements to produce per-isolate reports with acquired AMR genes, excluding common intrinsic AMR genes from being reported in specific species (Final AMR Gene Report), and phenotypic inference for specified species, where validated (Inferred Antibiogram Report, Fig. 2 and Supplementary Data 1). AMR, antimicrobial resistance.

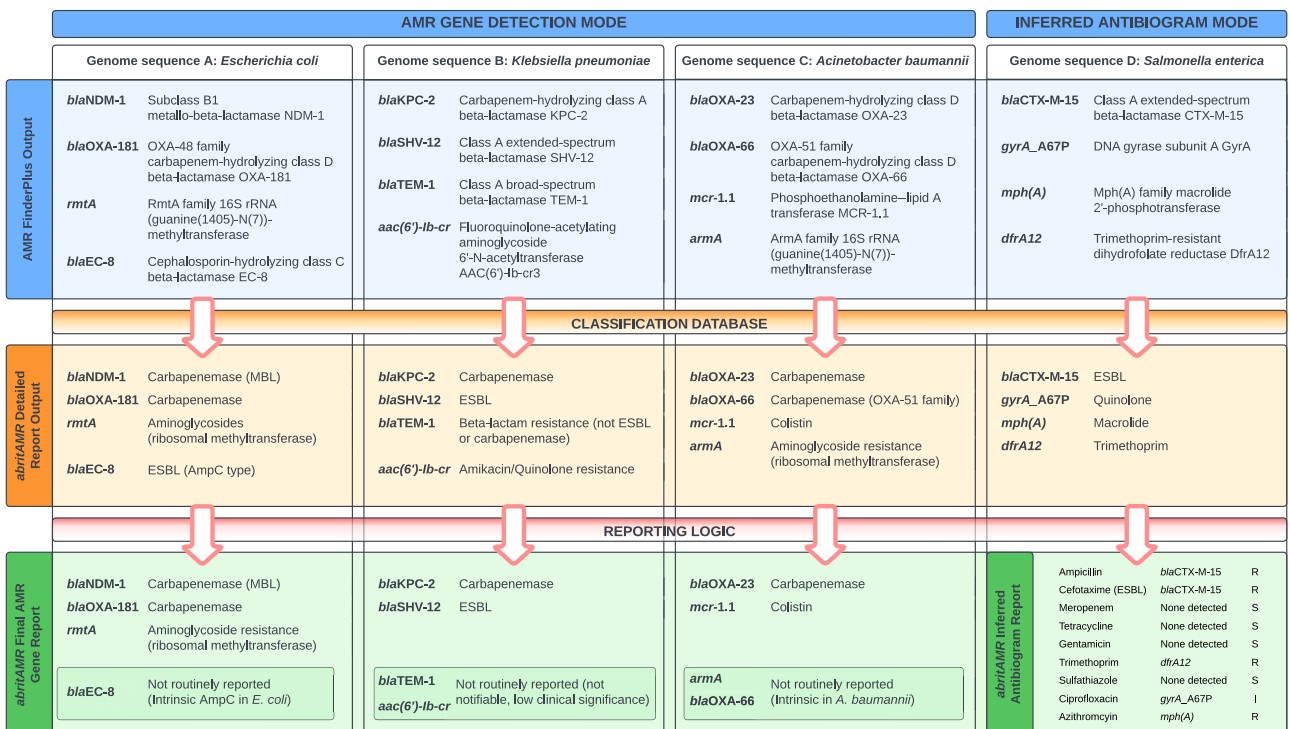

**Fig. 2 | Examples of *abritAMR* pipeline outputs.** This figure demonstrates *abritAMR* features and outputs across four different species (genome sequence A, *Escherichia coli;* genome sequence B, *Klebsiella pneumoniae;* genome sequence C, *Acinetobacter baumannii*; genome D, *Salmonella enterica*). The horizontal lanes represent the different output stages: top lane, raw *AMRFinderPlus* output; middle lane, *abritAMR* Detailed Report output (aligned to and binned by Enhanced Subclass from *abritAMR's* classification database); bottom lane, *abritAMR's* Final AMR Gene Report (genomes A-C) after application of tailored reporting logic to meet local requirements, and Inferred Antibiogram Report (genome D), currently validated for *Salmonella* spp. Key features include: (i) simplification of mechanism or drug class bins, (ii) identification and separation of high-priority AMR groups (such as notifiable AMR mechanisms) from lower-priority groups, (iii) identification of clinically-relevant AMR mechanisms within a drug class (e.g. separation of ESBLs and AmpCs from genes encoding first-generation cephalosporin resistance; separation of metallo-beta-lactamase (MBL) carbapenemases and other carbapenemases due to differences in patient treatment), and (iv) application of tailored reporting logic to separate reportable and non-reportable genes according to local requirements, thus de-cluttering the report for clinicians and public health teams. Note, only 'Exact' matches (100% sequence identity and coverage) and 'Close' matches (90-<100% identity and 90-<100% sequence coverage) from the *AMRFinderPlus* tool are reported by *abritAMR*. Abbreviations: MBL, metallo-beta-lactamase; ESBL, extended-spectrum beta-lactamase.

classes that are notifiable as part of our national critical antimicrobial resistance surveillance system (CARAlert) in Australia[17], with 99.9% accuracy (95% CI 99.9–100%), 98.9% sensitivity (98.3–99.3%) and 100% specificity (100–100%) across these classes (carbapenemases, 16 S ribosomal methyltransferases, mobile colistin resistance genes, ESBLs (including AmpCs), vancomycin resistance genes, and oxazolidinone and phenicol resistance (*optrA, cfr* and *poxtA* genes)).

**Validation results compared to PCR and Sanger sequencing**
The *abritAMR* pipeline was highly accurate compared to PCR (carbapenemase, ESBL, *van* and *mec* gene PCRs) with 1179/1184 (99.6%) resistance genes correctly detected, and compared well to Sanger sequencing (carbapenemase allele calling) with 355/356 (99.7%) alleles correctly identified by WGS. After discrepancy resolution (including repeat PCR and/or WGS, or examination of partial genes detected by *abritAMR*), five discrepancies between PCR and WGS results remained, including three potential false negatives (PCR positive, WGS negative) consisting of one CTX-M and two CMY genes not detected by *abritAMR;* at least one of these was due to the presence of a contig break in the gene leading to smaller fragments not detected by *AMRFinderPlus*. Additionally, two potential false positives (PCR negative, WGS positive) were identified, one CMY-42 and one IMP-62, confirmed by repeat PCR and sequencing; both genes were reported to be within the inclusivity range of the assay as per the manufacturer's instructions, although this was validated by in silico PCR by the manufacturer (and observed in our dataset). Alternatively, these discrepancies may be due to plasmid dropout in culture (which is commonly observed with suspected CPE isolates such as these), as different colonies with and without the ESBL/AmpC gene may have been picked for PCR and WGS, potentially explaining these discrepancies. No discrepant results were detected for *mecA* and *van* gene detection, and only one allele was incorrectly assigned compared to Sanger sequencing (99.7% accuracy) (Table 1 and Fig. 4). Overall performance of *abritAMR* against PCR yielded 99.6% accuracy (95% CI 99.0-99.9%), 99.6% sensitivity (99.0-99.9%) and 99.4% specificity (97.9-99.9%).

**Identification of AMR genes from synthetic reads**
The presence or absence of 415 AMR genes across 321 genomes (133215 alleles) was tested by running *abritAMR* on synthetic reads from complete reference genomes, and comparing to the (native) *AMRFinderPlus* results on the complete genome, considered the 'gold standard' (Fig. 3). Overall accuracy of AMR gene detection by *abritAMR* was excellent, with 133127/133215 alleles called correctly, resulting in 99.9% accuracy (95% CI 99.9-99.9%), 97.5% sensitivity (96.9-98.0%), and 100% sensitivity (100-100%) (Fig. 5). Note that any discrepancies here include differences in *abritAMR* performance compared to *AMRFinderPlus*, as well as differences between complete genomes and (synthetic) short-read data, which likely accounts for at least a proportion of the discrepant results.

The majority of discrepancies were false negatives, with the aminoglycoside AMR genes being most common (32/88, 36.4%), especially the *aac(6')-Ib* family, implicated in 18 false negatives, and specifically the *aac(6')-Ib-cr5* allele (11/18). Some of these were detected as partial genes at the site of contig breaks, possibly related to slightly higher GC

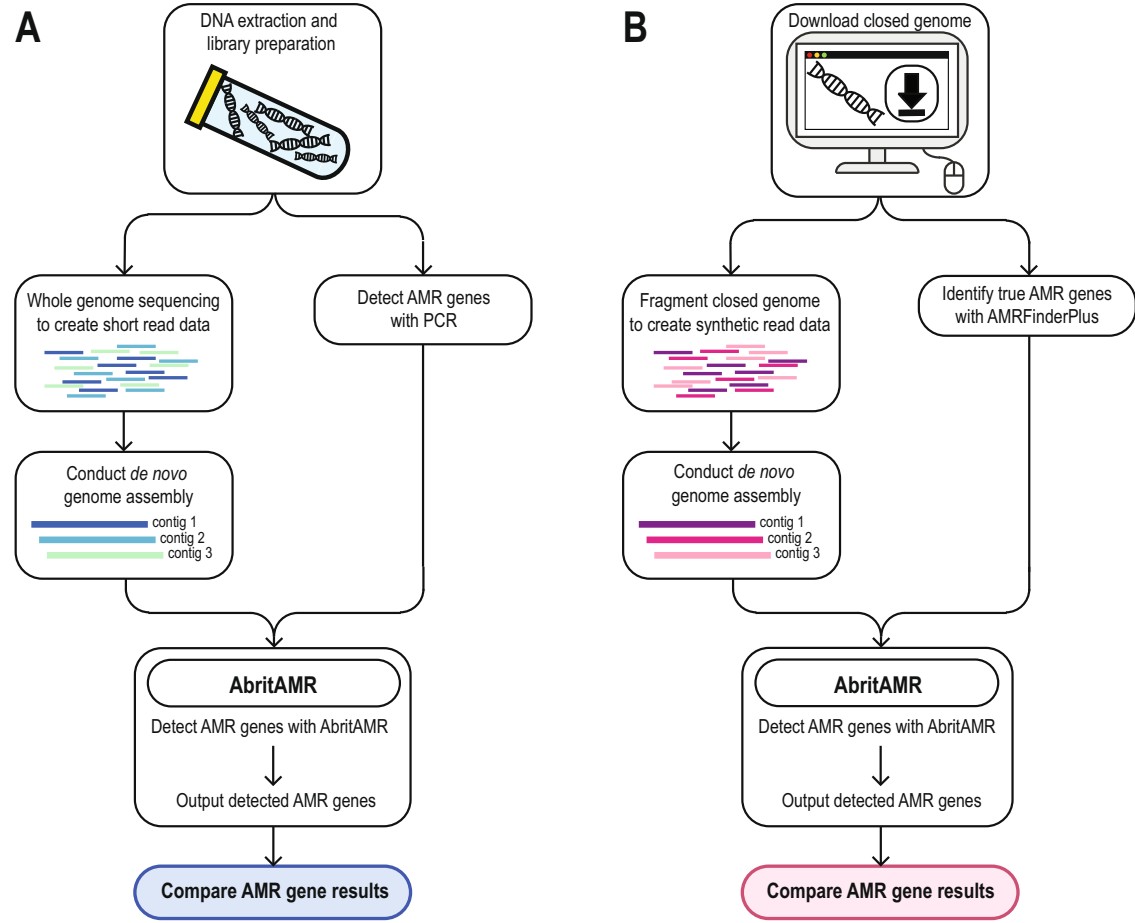

**Fig. 3 | Validation of *abritAMR* outputs compared to PCR and synthetic read data. A** Validation compared to PCR data – Assembled short-read sequence data from bacterial isolates are run through the *abritAMR* pipeline and compared to multiplex real-time PCR results for the same AMR genes. **B** Synthetic data – where no validation dataset or PCR assay was available for comparison, synthetic read data were generated from publicly-available closed reference genomes by fragmentation with the *art-illumina* tool[45] (using the error profile from local sequencing platforms) to mimic library preparation from bacterial DNA. Synthetic reads then underwent the same analytical processes as for usual usage (genome assembly and input into *abritAMR* pipeline). These results were then compared to *AMRFinderPlus* results on the complete bacterial reference genomes, minimizing the risk of discordant results due to disparities between AMR databases. Abbreviations: AMR, antimicrobial resistance; PCR, polymerase chain reaction. Source data are provided as a Source Data file.

content (leading to lower sequence coverage) in these genes. The other major theme was difficulty resolving sequences with multiple alleles of the same gene family, which were often collapsed into a single gene detection by *abritAMR* or miscalled as a different allele. For example, this included a sequence with CTX-M-3, CTX-M-14 and CTX-M-65 identified by *AMRFinderPlus*, and called as CTX-M-3 and CTX-M-24 by *abritAMR*. Four out of the five 'false positive' detections were actually allele miscalls within the same gene family. Collapse of repeated regions or duplicate alleles is often a feature of short-read sequencing, hence the discrepancies here may be a feature of comparing (synthetic) short-read data to complete genomes, rather than a feature of *abritAMR*. Notably, use of alternative genome assembly tools (SKESA[18] and SPAdes[19]) did not resolve the discrepancies, with similar performance to *Shovill*[20] (based on SPAdes; Supplementary Table 1).

### Limit of detection and precision

The limit of detection of the *abritAMR* pipeline was assessed to determine the minimum average sequencing depth for acceptable accuracy of AMR gene detection (as required for clinical microbiology validation and accreditation). Accuracy was found to be consistent (99.9%) across the 40X to 150X range, with 40X being the minimum coverage accepted by our accredited quality control (QC) pipeline. Repeatability and reproducibility (precision) were assessed (replicates within and across sequencing runs) and found to be 100% concordant.

### Validation of inferred antibiogram (*Salmonella* spp.)

Validation of inferred phenotype against phenotypic AST data demonstrated 98.9% accuracy (95% CI 98.7–99.1%), 98.9% sensitivity (98.4–99.3) and 98.9% specificity (98.7–99.1%) overall (Table 1, Supplementary Table 2 and Fig. 6). Accuracy of phenotypic inference was ≥98% for 11/13 antimicrobials (85%), with lower accuracy identified for streptomycin (95.5%, 95% CI 93.7–96.9%) and ciprofloxacin (96.8%, 95% CI 95.4–97.8%), similar to previous findings using different bioinformatic methods[21].

A number of 'false positive' results were identified for streptomycin (resistant genotype [AMR genes or mutations detected], susceptible phenotype; $n = 30/716$ (4.2%) isolates). The AMR genes detected in phenotypically susceptible isolates were also detected in non-susceptible isolates, although the non-susceptible isolates more often had >1 AMR gene (1 AMR gene, 22% phenotypically resistant; 2 or more AMR genes, 81.4% phenotypically resistant), suggesting that these AMR mechanisms had small but additive effects on phenotype. Evaluation of phenotype-genotype concordance for azithromycin identified five 'false negatives' (susceptible phenotype, no AMR mechanism detected) and two 'false positives' (AMR mechanism detected but phenotypically susceptible; neither isolate carried the dominant resistance mechanism for azithromycin in *Salmonella* spp. (*mph(A);* one carried *mef(B)*, an efflux pump with variable activity, and one carried *ere(A)*, an esterase with lower affinity for azithromycin[22]).

**Table 1 | Performance characteristics of *abritAMR* bioinformatic pipeline for detection of acquired AMR genes and inferred phenotype from WGS data**

| Validation panel | Accuracy (%, 95% CI) | Sensitivity (%, 95% CI) | Specificity (%, 95% CI) | PPV (%, 95% CI) | NPV (%, 95% CI) |
|---|---|---|---|---|---|
| Detection of acquired AMR genes | | | | | |
| PCR (n = 1184 isolates) | 99.6 (99.0–99.9) | 99.6 (99.0–99.9) | 99.4 (97.9–99.9) | 99.8 (99.1–100) | 99.1 (97.5–99.8) |
| Synthetic reads (n = 321 isolates, 133215 alleles[a,b]) | 99.9 (99.9–99.9) | 97.5 (96.9–98.0) | 100 (100–100) | 99.8 (99.6–100) | 99.9 (99.9–99.9) |
| Overall performance (PCR & synthetic data) | 99.9 (99.9–99.9) | 97.9 (97.5–98.4) | 100 (100–100) | 99.8 (99.6–99.9) | 99.9 (99.9–99.9) |
| Critical AMR subclasses[c] (all validation sets) | 99.9 (99.9–100) | 98.9 (98.3–99.3) | 100 (100–100) | 99.8 (99.5–100) | 100 (99.9–100) |
| Inferred phenotype concordance with AST (all antimicrobials)[d] | 98.9 (98.7–99.1) | 98.9 (98.4–99.3) | 98.9 (98.7–99.1) | 96.1 (95.2–96.8) | 99.7 (99.6–99.8) |
| Sanger sequencing for allelic variants (n = 356 alleles) | Accuracy 99.7% (355/356 alleles correctly identified) | | | | |
| Limit of detection | Accuracy 99.9% at all coverage levels tested (40X – 150X) | | | | |
| Precision (n = 13 isolates) | 100% repeatability and reproducibility (within- and between-run precision) | | | | |

'Detected' refers to AMR gene detected in reference dataset (PCR/allelic variant/synthetic reads) and also in *abritAMR* results from WGS.

'Not detected' refers to AMR genes detected in reference dataset, but not WGS.

CI confidence interval, PPV positive predictive value, NPV negative predictive value, WGS whole genome sequencing, AMR antimicrobial resistance, AST antimicrobial susceptibility testing; X, times.

[a]Presence or absence of each of 415 AMR gene alleles was assessed across the dataset.

[b]Presence or absence of each of 415 AMR gene alleles was assessed across the 321 genomes, resulting in 133215 alleles for comparison across the dataset.

[c]Calculated by drug class (enhanced subclass from classification database). Range of sensitivity varied from 70-100% for each subclass (see Supplementary Table 3 for subclasses, and Fig. 5 for performance across subclasses).

[c]Critical AMR subclasses – defined as classes containing AMR genes nationally-reportable to the CARAlert program. Includes all carbapenemases, ESBL, ESBL (AmpC type), ribosomal methyltransferases, colistin, oxazolidinone & phenicol resistance, vancomycin.

[d]Inferred phenotype validation for *Salmonella* spp., compared to agar dilution (CLSI methods and 2020 breakpoints). Detailed performance metrics available in Supplementary Table 2 and Fig. 6.

Similar to streptomycin, ciprofloxacin also had a number of phenotype-genotype mismatches, likely due to low-level resistance conferred by AMR mechanisms. Isolates with one AMR gene most often had an intermediate phenotype (81.3% intermediate, 12.4% resistant, 6.2% susceptible), whilst isolates with ≥2 AMR genes were all phenotypically resistant. Despite these discordances, the best correlation between genotype and phenotype (S/I/R) was determined according to the number of AMR mechanisms detected (any type). This was coded into the reporting logic: absence of AMR mechanisms, 'susceptible', one AMR mechanism, 'intermediate', two or more AMR mechanisms, 'resistant'. In this application, the use of an 'intermediate' category implies that MICs are likely to be borderline for these samples, i.e. may test susceptible or resistant on AST. Note that these results are only used for epidemiologic purposes (not for patient treatment), and hence over-calling resistance is more suitable for this purpose than non-detection of AMR mechanisms.

### Sample outputs and incorporation into microbiology report

Two different outputs are used in the *abritAMR* pipeline: (i) Detailed Report output, where AMR genes or mutations are shown by enhanced subclass, as classified by the *abritAMR* database, and (ii) Final AMR Gene Report output (binned into reportable and non-reportable genes) after reporting logic is applied (Fig. 2 and Supplementary Data 1). An example of the output of the additional module incorporating mutational resistance and Inferred Antibiogram Report phenotype (currently validated for *Salmonella* spp.) is also shown in Supplementary Data 1.

### Implementation processes

After the validation process, implementation processes included modifying report outputs, integration with the existing LIMS, and documentation of the standard operating procedure (SOP). Multiple groups were consulted on the proposed report outputs, including reporting scientists with domain expertise (ensure results were easily interpretable and met reporting obligations across different pathogens), quality management staff (ensure results met legal requirements), and end-users (public health teams and clinicians). Subsequently, all staff involved in detection, reporting or interpretation of AMR results were trained in the use and interpretation of *abritAMR*, and clients were educated about the change (although only minimal differences were noticeable to clients, such as change in report formats) before full implementation into routine workflows. Implementation led to streamlined workflows, including rapid bioinformatic processing of large sequencing runs (AMR gene detection for a 96-sample run completed in <3 min with 256 CPUs), and less manual re-classification of AMR gene results by laboratory scientists (e.g. moving genes between reportable and non-reportable fields, removing intrinsic AMR genes from reportable fields).

## Discussion

The use of genomics in clinical and public health microbiology (CPHM) has increased substantially in the last decade, particularly in the fields of pathogen typing and outbreak investigations[5,7,23]. Detection of AMR from WGS data has somewhat lagged behind other applications of WGS, likely due its inherent complexity in comparison to simple and effective phenotypic AST[4]. This complexity is multi-faceted but includes the vast array of resistance mechanisms for testing (a single phenotype may be encoded by many different AMR mechanisms), and the limitations of phenotype-genotype correlation, particularly for less-common organisms and drug classes[24,25]. If not addressed systematically, these issues may render genomic AMR difficult to identify comprehensively across all pathogens seen in a CPHM laboratory, and difficult to communicate to clinicians and public health units[13,26].

Globally, the paucity of highly accurate, reproducible bioinformatic tools for detection of AMR mechanisms has been recognised as

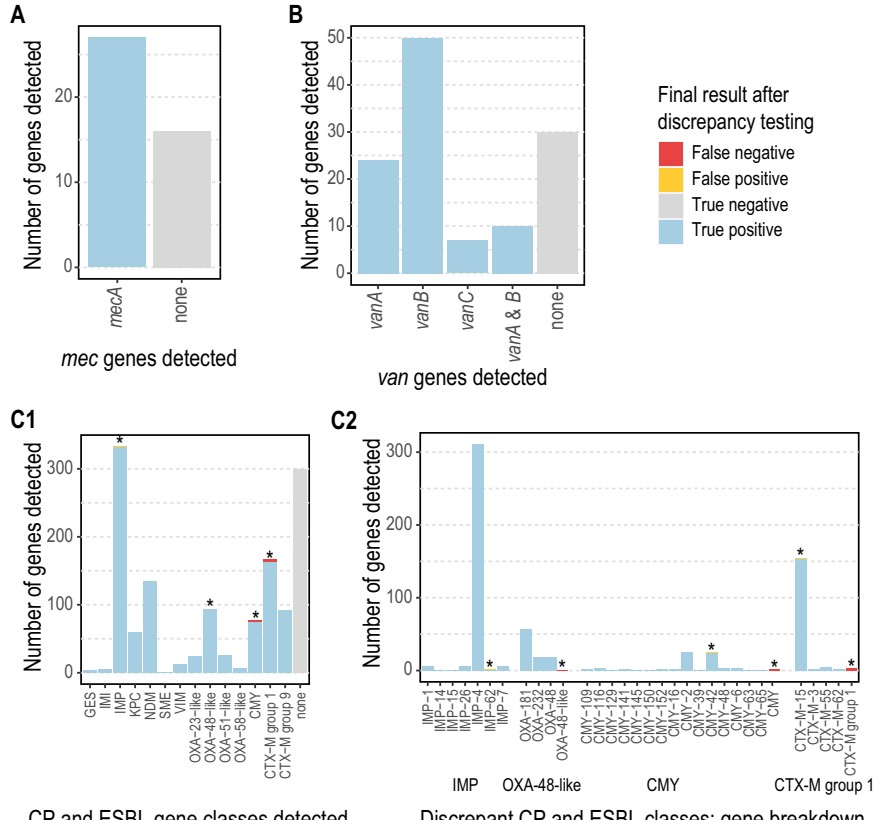

**Fig. 4 | Performance of *abritAMR* pipeline to detect AMR mechanisms compared to PCR.** Each panel details the identification of AMR mechanisms by *abritAMR* compared to the 'gold standard' multiplex PCR assays used in our laboratory. True positive, detected by both PCR and *abritAMR*; true negative, not detected by either PCR or *abritAMR*; false positive, detected by *abritAMR* but not multiplex PCR, *and* within the known range of the PCR assay; false negative, detected by multiplex PCR but not by *abritAMR*. Panel **A**, *mec* genes compared to multiplex PCR (*mecA/mecC*, no *mecC* detected by either method). Panel **B**, *van* genes compared to multiplex PCR (*vanA/B/C*). Panel **C1**, detection of genes within carbapenemase and ESBL gene families compared to multiplex PCR panel (AusDiagnostics CRE panel); asterisks represent groups where discrepancies were identified, and expanded out in Panel **C2** to show the specific gene discrepancies between the two methods. Abbreviations: AMR antimicrobial resistance, PCR polymerase chain reaction, CP carbapenemase, ESBL extended-spectrum beta-lactamase. Source data are provided as a Source Data file.

one of the main limiting factors to wider application of genomics in the CPHM setting[4,27]. Here, we have designed and validated a bioinformatic platform for genomic detection of AMR determinants across bacterial species focusing on reporting requirements for clinical and public health microbiology, performed a rigorous validation, and implemented it to achieve an ISO-ccertified genomic workflow for AMR. This was achieved by adapting an existing software tool and database (*AMRFinderPlus*), and adding a modified classification step plus reporting logic to produce tailored reports for a CPHM audience.

This platform relies heavily on the comprehensive, well-curated and frequently updated AMR database behind *AMRFinderPlus*, as well as the excellent software tool, which uses multiple search methods to best identify AMR genes and mutations (with results annotated by the type of 'match', to allow scientists and clinicians to understand the degree of confidence behind each call)[16]. Notably, outputs for other large AMR databases, such as CARD[28] and ResFinder[29], could be modified to achieve similar tailored reports to *abritAMR*; our choice was based on the ease of integration into our existing workflows and reporting. *abritAMR's* speed allows for rapid detection of AMR genes in routine high-throughput workflows, with AMR gene detection completed on a 96-sample sequencing run within 3 min. The addition of mutations to *AMRFinderPlus* for an increasing number of species has been very useful in our early applications, enabling our public health laboratory to move to a fully-genomic workflow for *Salmonella* surveillance, as all samples were sequenced for typing and phylogenetic analysis (a sub-sample of isolates still undergo AST to ensure new AMR

mechanisms are detected). The capacity to include AMR genes or mutations of local significance would be a welcome addition to *AMRFinderPlus*, further extending its utility.

When this work commenced, there were no classifications of AMR mechanisms into drug classes in the *AMRFinderPlus* database, hence we created our own classification database, which has also evolved in parallel with the great advances made by the *AMRFinderPlus* team. There are (now) a small number of essential differences in the drug class classifications that we feel are important to enhance its utility for CPHM. Key examples include separating carbapenem resistance into different groups based on their mechanisms; separating into 'carbapenemase', 'carbapenemase (MBL)' and 'carbapenemase (OXA-51 family)' enables reporting each group separately, as antibiotic choices differ with metallo-beta-lactamases (MBLs) compared to non-MBL carbapenemases, and OXA-51 family carbapenemases are weak and intrinsic to *Acinetobacter* spp. and routine reporting is not required (coded as part of reporting logic). This combination of tailored classification and reporting logic allows the vast and complex array of AMR mechanisms to be distilled into results and reports that can be understood by scientists, clinicians and public health teams alike, without a great deal of prior knowledge. Future development will focus on restructuring the database to include different levels (classes, subclasses) to take advantage of the higher resolution of classifications now included in *AMRFinderPlus*.

Notably, the key limitations of *AMRFinderPlus* are also limitations of both ResFinder[30] and CARD-RGI[31] tools in different

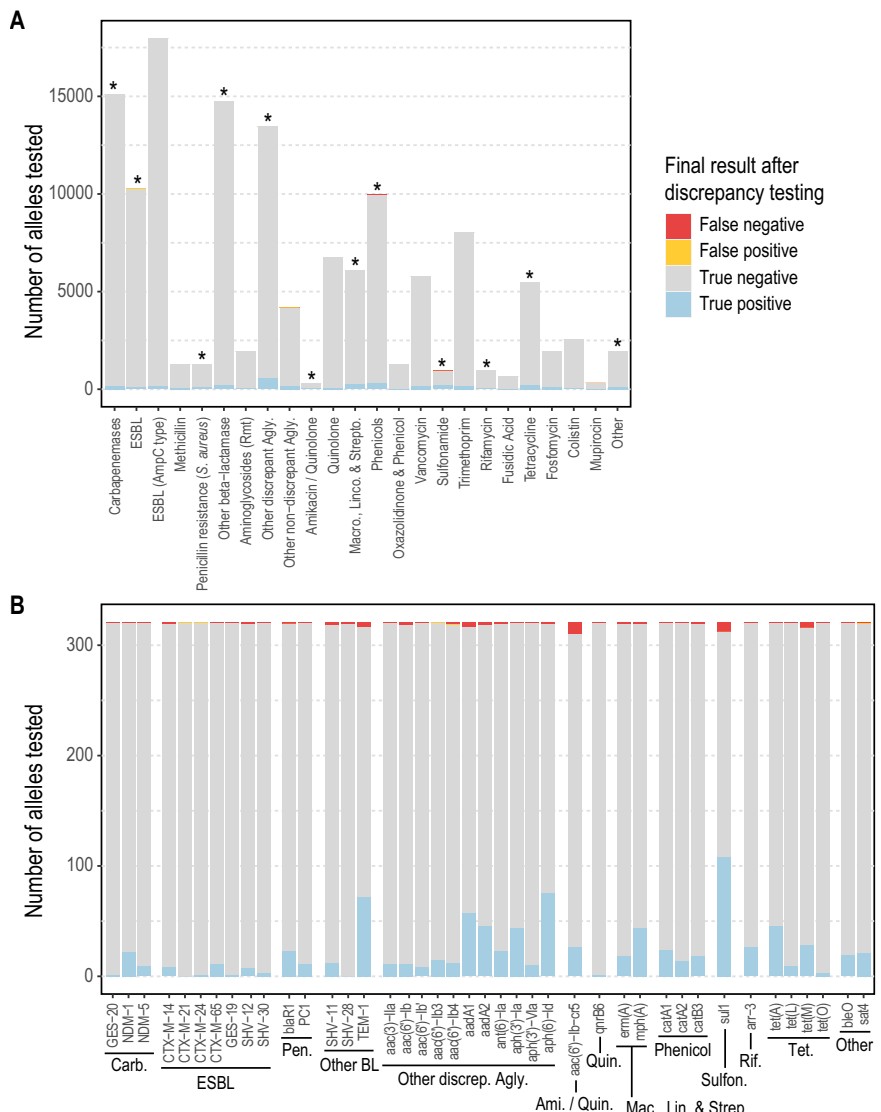

**Fig. 5 | Performance of *abritAMR* pipeline compared to synthetic read data.** This figure shows the presence or absence of 415 AMR genes across 321 genomes identified by *abritAMR* (performed on assembled synthetic read data) compared to *AMRFinderPlus* on complete genome sequences ('gold standard'). True positive, detected by *AMRFinderPlus* and *abritAMR*; true negative, not detected by either *AMRFinderPlus* or *abritAMR*; false positive, detected by *abritAMR* but not by *AMR-FinderPlus*; false negative, detected by *AMRFinderPlus* but not by *abritAMR*. Panel **A**, *abritAMR* results by enhanced subclasses (further grouped to simplify visualisation); asterisks represent classes with any discrepant result (false positive or false negative), and are examined in more detail in Panel **B**. Panel **B** shows a detailed view of genes with discrepant results within each subclass. A full list of AMR subclasses included in the validation set can be found in Supplementary Table 3. Abbreviations: EBL extended-spectrum beta-lactamase, Rmt ribosomal methyltransferase, Agly aminoglycoside, Macro., Linco. & Strepto, macrolides, lincosamides and streptogramins (combined class); Carb., carbapenemase; Pen., penicillin resistance (*S. aureus*); Other BL, other beta-lactamase; Other discrep. Agly, other discrepant aminoglycoside subclass; Ami./Quin., amikacin/quinolone subclass; Quin., quinolones; Mac., Lin. & Strep., macrolides, lincosamides and streptogramins; Sulfon., sulfonamides; Rif., rifampicin; Tet., tetracyclines. Source data are provided as a Source Data file.

ways; ResFinder classifies AMR determinants into a small number of antimicrobial classes, but lacks the resolution needed for the more difficult classes described above (particularly beta-lactams), whilst CARD-RGI maintains an ontologic focus, where antimicrobial targets and mechanisms are identified at varying levels, but not grouped in a way that facilitates CPHM reporting and clinician understanding. However, both tools offer the accessibility of a graphical user interface (GUI) and the option of using raw reads as inputs for analysis, which are particularly important considerations for laboratories without dedicated bioinformatic expertise. Ideally, all large AMR databases and tools should facilitate clinically-relevant reporting of AMR determinants through 'interpretation' of outputs for clinical needs, and modifiable reporting logic to tailor outputs to reporting requirements, whilst maintaining a balance between accessibility and accuracy to enable validation and accreditation.

In a CPHM setting, it is critical to validate any new test or analytical process to ensure the veracity of results, and that the results (outputs or reports in this case) are fit-for-purpose[12]. However, formal test validation and accreditation procedures are based on wet-lab assays, and not always easily transferable to new methods such as WGS[32]. This may require some creative thinking about different ways to validate a new genomic test[33], as demonstrated here with the use of synthetic sequencing reads generated from complete reference genomes. Ideally, a broad range of publicly-available reference datasets with genotypic and phenotypic data would be made freely available to assist with validation and bench-marking for databases, tools and new pipelines such as this, greatly advancing the development of AMR genomics[4,34].

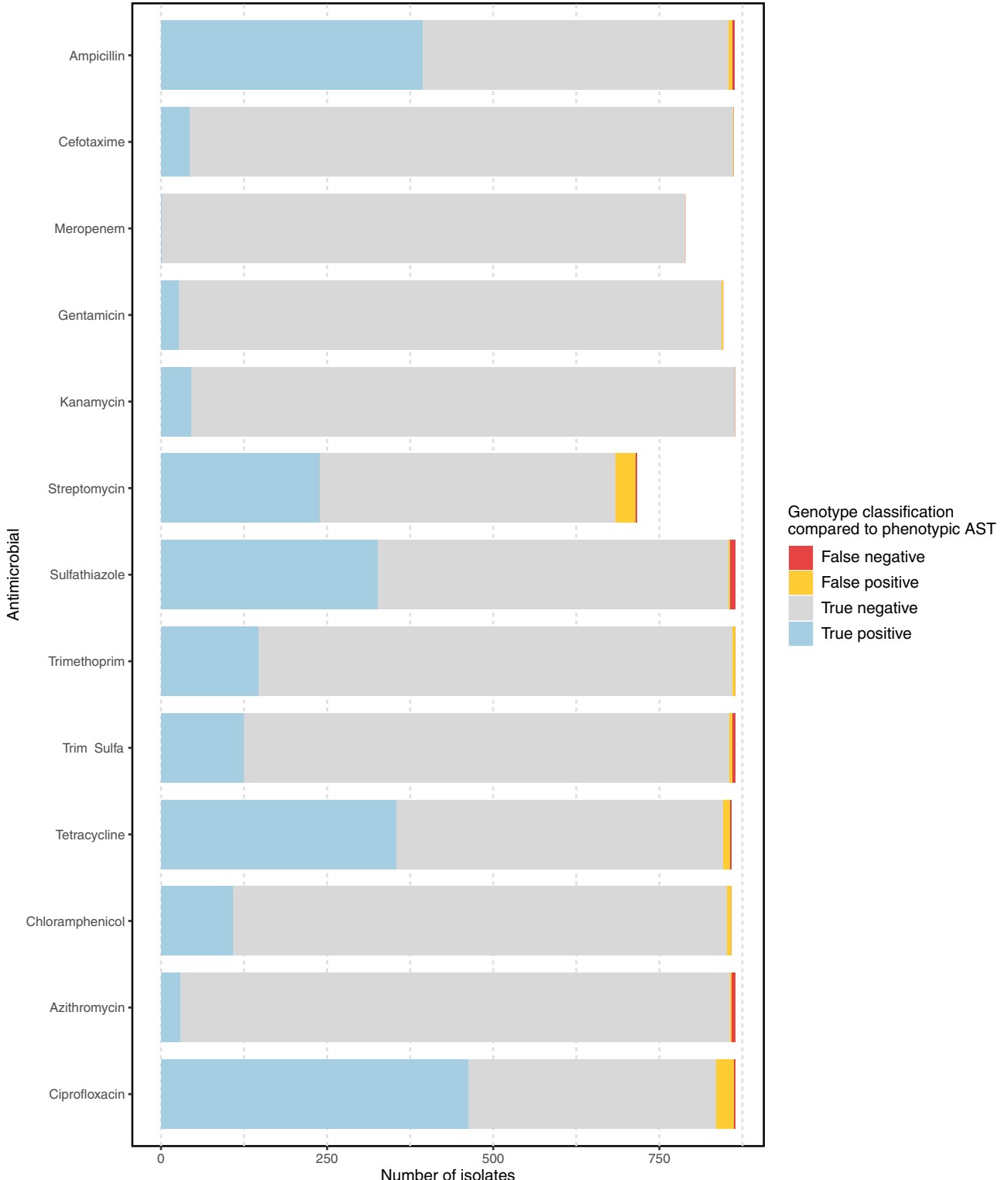

**Fig. 6 | Performance of inferred phenotype from *abritAMR* compared to antimicrobial susceptibility testing (AST).** Classification of genotype (AMR mechanism detection) compared to the 'gold standard' phenotypic AST for each isolate and antimicrobial. True positive, genotypic and phenotypic resistance; true negative, no AMR mechanisms detected in phenotypically susceptible isolate; false positive, AMR mechanism identified in phenotypically susceptible isolate; false negative, no AMR mechanisms detected in phenotypically resistant isolate. For ciprofloxacin, 'true positive' defined as concordant intermediate or resistant results (phenotype and genotype). Abbreviations: AST, antimicrobial susceptibility testing; AMR, antimicrobial resistance; Trim-sulfa, trimethoprim-sulfamethoxazole. Source data are provided as a Source Data file.

Initiatives such as the NCBI National Database of Antibiotic Resistant Organisms (NDARO)[35] and PATRIC[36] are promising, but currently limited in scale, and further global data sharing is required here to advance phenotype-genotype correlations. Here, we have contributed a dataset that may be used for validation of genomic detection of AMR determinants against PCR results, and a method for validating against synthetic genomic data, to assist other laboratories to validate their own AMR workflows.

As with all attempts to validate new WGS pipelines and workflows, our study has limitations. The absence of a 'gold-standard' dataset to compare results from our pipeline to means that we must compare to imperfect standards, such as existing testing methods with lower resolution (PCR) and use synthetic sequencing data to compare targets not covered by PCR in our laboratory. Until these issues are addressed globally, laboratories will have to persist with these challenging comparisons, and rely on new initiatives such as proficiency testing programs (PTPs) for WGS (with participation being a requirement for test accreditation in our setting) to start to standardise results across laboratories and countries. Whilst *abritAMR* was highly accurate overall, a small proportion of discrepant results were identified, of which the majority were false negative results. Most of these discrepancies are likely due to the comparison of synthetic short-read data to complete genomes, where contig breaks within a gene result in non-detection, or plasmid dropout in culture, leading to non-identification of plasmid-borne AMR determinants in WGS data. In our validation, use of a different genome assembly tool had minimal impact, although it is critical to include this as a consideration in the validation process, particularly where discrepancies are present.

We envisage that the *abritAMR* pipeline will most likely be applied in CPHM settings, and hope that it may assist sequencing laboratories address the difficult question of how to best report these data to clinicians and public health teams with limited AMR knowledge. However, it may also have utility in other settings including research, particularly where complex AMR data need to be binned into functional classes to facilitate understanding when the user is less familiar with AMR. In our view, it is critical for medical microbiologists, scientists and bioinformaticians to continue to work together to navigate the challenges of communicating complex AMR data to clients, to advance the reach of genomic AMR and maximise the benefits of this potentially transformative technology.

## Methods

### Setting and existing genomics workflow

The Microbiological Diagnostic Unit Public Health Laboratory (MDU PHL) is a state reference laboratory for bacterial pathogens, including carbapenemase-producing *Enterobacterales* (CPE), *Acinetobacter* spp., *Pseudomonas* spp., vancomycin-resistant enterococci (VRE) and enteric pathogens[37–39]. The laboratory has a strong emphasis on genomics, primarily for epidemiologic surveillance, with increasing applications for clinical purposes. In conjunction with the Department of Health Victoria, we have embarked upon a broad program to increase implementation of pathogen genomics for public health purposes, either enhancing or superseding current laboratory methods.

Our existing genomics workflow (incorporating sample receipt, nucleic acid extraction, library preparation, short-read sequencing (Illumina NextSeq or MiSeq), and quality control (QC) of reads including de novo genome assembly) has already been validated and accredited by the National Association of Testing Authorities Australia (NATA, analogous to Clinical Laboratory Improvement Amendments [CLIA] in USA)[40–42]. Details of this workflow can be found in Supplementary Methods. Briefly, single colonies from overnight pure bacterial sub-cultures were selected and placed in lysis buffer. DNA extraction was performed on the QIAsymphony using the DSP Virus/Pathogen Mini Kit, and library preparation performed using Nextera XT (Illumina Inc.) according to manufacturer's instructions. WGS was performed on NextSeq 500/550 or MiSeq platforms (Illumina Inc.), generating 150 bp or 300 bp paired-end reads respectively. Reads were assembled de novo using *Shovill*[20]. QC requirements for fastq reads to be included in subsequent analysis were (i) Q-score ≥30, (ii) data with a minimum estimated average genome coverage of >40X, and (iii) estimated genome size within range for observed species (see Supplementary Methods for detailed descriptions).

### The *abritAMR* bioinformatics pipeline

The aims for development of this bioinformatic pipeline were to detect AMR genes and mutations accurately and reliably from bacterial whole genome sequencing (WGS) data, which could be validated against PCR and other data sources, implemented in a public health or clinical microbiology laboratory, and successfully accredited by governing bodies. The *abritAMR* bioinformatic platform takes a genome assembly from short-read data, long-read data or hybrid assemblies (fasta file) as input (once it has met defined QC parameters), and includes five main components (Fig. 1):

i. NCBI's *AMRFinderPlus* tool (https://github.com/ncbi/amr) – *abritAMR* implements this tool to identify AMR genes in genome sequences, using a combination of BLASTx (matching the protein sequences of AMR genes to the protein sequence of the query isolate) and Hidden Markov Models (HMMs)[16].

ii. NCBI's *AMRFinderPlus* database – *abritAMR* uses this frequently updated database (https://github.com/ncbi/amr/wiki/AMRFinderPlus-database), which is a comprehensive and extensively curated database of AMR gene sequences. Current functionality includes mainly AMR genes ('core' database), with point mutations (species-specific) and virulence genes increasingly being included in the 'plus' database. In more recent iterations, AMR genes and point mutations include information about the antimicrobial class and subclass (or specific antimicrobials) that they confer resistance to.

iii. Classification database – While the *AMRFinderPlus* database includes some information about the antibiotic class and subclass affected for each AMR gene, these classifications are not always easily translatable for clinical and public health practice. For example, the beta-lactam subclass 'cephalosporin' includes AMR genes conferring resistance to first-generation cephalosporins (narrow-spectrum cephalosporinases, such as $bla_{OXA-1}$), or third-generation cephalosporins (such as $bla_{CTX-M}$ ESBLs), which have very different implications for AMR surveillance and patient management. The local *abritAMR* classification database is based on the current version of the *AMRFinderPlus* database, with an added field ('Enhanced subclass') to translate the NCBI subclasses into more functional versions for our purposes (logic detailed in Supplementary Table 3, examples in Fig. 2). This field is updated following each new database release (logic detailed in Supplementary Table 4).

iv. Species-specific reporting logic (AMR genes, all species) – Currently, most AMR genes detected by this pipeline are not required to be reported for surveillance or clinical purposes; reporting data on *all* AMR genes found in an isolate runs the risk of overwhelming clients with unnecessary data and missing the most pertinent AMR genes detected. As such, we developed a reporting logic process to filter the AMR genes detected in each isolate into 'reportable' or 'non-reportable' categories, to mirror the usual reporting requirements diagnostic laboratories (Supplementary Figure 1). This logic takes into account the species when determining what is reportable, limiting the reporting of intrinsic AMR genes (such as $bla_{OXA-51}$ subtypes in *Acinetobacter baumannii*), and differentiating between AMR genes that are only reportable in certain species (e.g. ESBL genes reportable for national surveillance of *Salmonella* spp.), while always reporting significant AMR genes that are not limited by species (e.g. carbapenemase and *mcr* genes). Non-reportable genes are also made available to the reporting pathologists and senior scientists and recorded in the laboratory information management system (LIMS), enabling detailed review of all detected AMR genes, correlation with phenotype, and movement between reportable and non-reportable categories when required as part of any routine results review process before reporting.

v.  Inferred phenotype (AMR genes and mutations, validated species only) – The pattern of AMR genes and mutations detected can be used to infer phenotype for a given isolate. In *abritAMR*, this is currently validated for *Salmonella* spp., and reported for epidemiologic purposes in our laboratory, replacing routine antimicrobial susceptibility testing (AST) of *Salmonella* spp. for public health surveillance (reporting logic detailed in Supplementary Figure 2, Inferred Antibiogram Report example shown in Supplementary Data 1).

### *abritAMR* outputs

*abritAMR* outputs include a Detailed Report output, consisting of a table (comma separated values file) of AMR genes or mutations detected for each sample, listed by enhanced subclass (e.g. "Carbapenemase (MBL)", "Colistin"), or a Final AMR Gene Reports, a table of AMR genes detected for each sample binned into 'reportable' or 'not reportable' fields, when the species-specific reporting logic is applied (Fig. 2). Additionally, when run on validated species (currently *Salmonella* spp.), *abritAMR* also produces an Inferred Antibiogram Report. All alleles listed in these outputs are either 'exact matches' (100% identity and 100% sequence coverage compared to the reference protein sequence) or 'close matches' (90-<100% identity and 90-<100% sequence coverage compared to the reference protein sequence, marked by an asterisk [*] to distinguish from exact matches), as defined by *AMRFinderPlus*. Partial matches (>90% identity, 50-<90% coverage compared to reference protein sequence) are listed separately, and must be examined further if deemed suitable for reporting. Where an internal stop codon (i.e. truncated gene) or HMM match are recorded by *AMRFinderPlus*, no result is reported by *abriTAMR*. Examples of *abritAMR* pipeline outputs are shown in Fig. 2, demonstrating how the *AMRFinderPlus* output is modified by *abritAMR* (binned into enhanced subclass according to *abritAMR's* classification database, and separated into reportable and non-reportable categories by the reporting logic).

### Validation of the *abritAMR* pipeline

To validate *abritAMR*, results from the pipeline were compared to results from PCR testing, Sanger sequencing and synthetic read sets as detailed below. For the purposes of validation, both 'exact' and 'close' matches were considered as 'detected'. Pre-specified sensitivity and specificity thresholds were defined for successful validation prior to analysis.

### Validation datasets

All isolates used in validation were obtained as part of routine AMR surveillance under public health laboratory functions, and hence were exempt from requiring ethics approval. Data were de-identified for the validation study (no patient or clinical data were used).

**PCR.** This dataset included 1184 bacterial isolates (42 species), that had previously been tested by PCR, including a carbapenemase and ESBL real-time multiplex PCR (*n* = 1020 isolates, AusDiagnostics 16-well CRE panel, catalogue no. 21098, version 03; Sydney, Australia), *van* gene PCR (*n* = 121, in-house assay for *vanA*, *vanB*, *vanC1* and *vanC2/3* genes[43]) and *mecA* PCR (*n* = 43, in-house assay for *mecA*[44])(Supplementary Figures 3–6).

**PCR and Sanger sequencing for allelic variants.** This dataset included 347 isolates (20 species) with carbapenemase resistance genes detected by a range of carbapenemase and ESBL PCR assays across six different carbapenemase resistance gene families (targets and primers detailed in Supplementary Table 5), with Sanger sequencing subsequently performed to identify the carbapenemase allelic variant (Supplementary Figures 7 & 8).

**Synthetic reads.** For the remaining AMR gene targets where PCR was not readily available to compare with *abritAMR*, we created synthetic short-read sequence data from complete, publicly available genomes from RefSeq or GenBank, and compared *abritAMR* results on synthetic short reads to *AMRFinderPlus* results from the complete genomes. To do this, we generated synthetic 150 bp paired-end reads using the *art-illumina* tool[45] to fragment the complete genome sequences, incorporating error profile data from a NextSeq500 sequencer, at 40X to 150X average genome coverage (40X is the minimum coverage accepted for QC) (Fig. 3). This dataset comprised 321 isolates (49 species) covering 415 unique AMR alleles from 43 resistance subclasses (Supplementary Table 3 and Figures 9 & 10). *abritAMR* results from synthetic reads were compared to (native) *AMRFinderPlus* results from complete genome sequences. This allowed direct comparisons of presence or absence of AMR genes, therefore avoiding the problem of discrepancies in AMR gene nomenclature that may lead to false discordance if two different AMR gene databases were compared.

**Precision testing.** *abritAMR* results from a test panel of 13 organisms (12 genera, Supplementary Table 6) sequenced multiple times (both within and across sequencing runs) using different sequencing platforms in our laboratory (NextSeq and MiSeq) and a range of sequencing modes (low, mid and high throughput) and read lengths (75–300 bp). Different combinations were compared to assess analytical precision (repeatability and reproducibility).

**Determination of limit of detection.** The limit of detection for molecular assays is normally the lowest amount of nucleic acid target that can be detected by the assay. This definition is not strictly applicable to whole genome sequencing, as the WGS assay is qualitative with a standardised DNA concentration being used in the sequencing reaction. Instead, the limit of detection in this context was calculated as the minimum average coverage across the genome required for accurate detection of gene targets or allele variants. Synthetic paired-end reads (150 bp) were generated at a range of sequencing coverages, from the minimum average coverage accepted for our routine QC (40 X) up to 150 X coverage.

**Determination of inferred phenotype (*Salmonella* spp.).** We validated phenotypic inference (Susceptible/Intermediate/Resistant, S/I/R) against an existing dataset of 864 sequenced *Salmonella* spp. with antimicrobial susceptibility (AST) data generated by agar dilution from 2018-2019. For the fluoroquinolone drug class, the S/I/R phenotypes associated with combinations of AMR genes and mutations were analysed to determine the relative weighting of each AMR mechanism to infer a phenotype most reliably from in silico analysis.

### Discordant result resolution

Discordant results were divided into two categories: firstly, PCR negative, WGS positive (false positive) - this may be due to the AMR gene detected by WGS not being included in the range of the PCR panel. If the gene was known to be included in the range of the PCR panel (as stated by the manufacturer), the isolate was retested by PCR and WGS to resolve this discrepancy. Secondly, PCR positive, WGS negative (false negative) – this may be due to an AMR gene being fragmented across two or more contigs, hence partial matches were assessed; if no partial matches were found, the sequence was interrogated using alternative tools; if this failed to resolve the discrepancy, the isolate was retested by PCR and WGS. Where possible, discrepancies between phenotypic and genotypic results were investigated through repeat phenotypic testing and/or repeat sequencing of the isolate.

### Re-verification processes

In accordance with ISO standards, the *abritAMR* pipeline must be re-verified after each database or tool update. Database updates are

reverified by confirming that the updated database performs to the same criteria as was defined in the original validation, using the synthetic dataset described above ('*abritAMR* test suite'). Updates to the *abritAMR* software may take the form of minor patches or major updates. Minor patches are changes that do not impact underlying structure or core logic of the pipeline, such as fixes for typographical errors or addition of functionality which does not impact the core logic of the tool, e.g. changes to log outputs. In these cases, a full reverification is deemed unnecessary and running of the *abritAMR* test suite is sufficient. However, other changes which may impact the core logic or structure of the outputs require a complete reverification as described for database updates. Any change in performance is assessed, the cause identified, and modifications made before the changes are implemented for reporting. All changes to *abritAMR* are tracked in GitHub and the versions managed using conda.

## Statistical analysis

Test performance characteristics (accuracy, sensitivity, specificity, positive and negative predictive values, including confidence intervals) were calculated using the epiR package for R (version 4.1.1), used in RStudio (version 1.4.1717).

## Reporting summary

Further information on research design is available in the Nature Portfolio Reporting Summary linked to this article.

## Data availability

Sequence data used in this study are available on NCBI Sequence Read Archive (BioProjects PRJNA529744, PRJNA565795, PRJNA856406, PRJNA856415, PRJNA857525, PRJNA857526, PRJNA857528, PRJNA857531, PRJNA857533, PRJNA857534, PRJNA870170 and PRJNA319593) with accession numbers provided in Supplementary Data 2. Accession numbers for the complete genomes used to generate the synthetic validation dataset are provided in Supplementary Data 2. PCR results for the PCR validation dataset are available in Supplementary Data 2 and on GitHub (https://github.com/MDU-PHL/abritAMR)[46]. Source data are provided with this paper.

## Code availability

Code for the *abritAMR* pipeline is publicly available at https://github.com/MDU-PHL/abritAMR (https://doi.org/10.5281/zenodo.7370627).

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

## Acknowledgements

MDU PHL is funded by the Victorian Government Department of Health. BPH receives an investigator grant from National Health and Medical Research Council Australia (GNT1196103). NLS received an Australian Government Research Training Program (RTP) scholarship. We sincerely thank the NCBI's *AMRFinderPlus* team for their dedication to producing and maintaining high-quality tools and database for detection of AMR mechanisms from WGS data. We also thank Cheryll Sia for sharing her insights on genotype-phenotype correlations for fluoroquinolone resistance in *Salmonella*.

## Author contributions

N.L.S., K.H., B.P.H., A.G.S. and T.S. conceived the project. N.L.S. and K.H. designed the software with input from A.G.S., T.S. and M.B.S. N.L.S. and K.H. wrote the manuscript, C.L.G. created figures, B.P.H., T.P.S. and T.S. supervised the manuscript writing and editing. K.H., T.S., S.A.B., N.L.S. and A.G.S. designed the validation. M.L.S., K.S. and M.V. contributed to the validation, N.L.S. and K.H. performed and analysed the validation results. All authors reviewed, edited and approved the manuscript.

## Competing interests

The authors declare no competing interests.
