## [Peer Review File · Nature Communications]

An ISO-certified genomics workflow for identification and surveillance of antimicrobial resistanceReviewer #1 (Remarks to the Author):

This is a very well-written, but also quite long manuscript. I am personally a bit uncertain regarding the scientific novelty of the work.

On one hand it is certainly admirable that the authors try to promote and write about bioinformatics pipelines for routine diagnostics. This requires a lot of work and it is often very difficult to get acknowledged in the scientific literature compared to for example descriptions of yet another outbreak. However, implementing WGS based ISO-accreditation is not novel. This has been done in several laboratories globally and has been running in some places for 5-6 years (Genome Trakr, Genomics England, etc.). On that account, I also miss that the authors at least acknowledge the ISO-standards within this area (<https://www.iso.org/obp/ui/#iso:std:iso:23418:ed-1:v1:en>).

Regarding the tool itself, I do believe the authors are overselling it a bit. As far as I can judge, this works on assemblies only and require command-line expertise. The proposed pipeline is easy to install and use for a bioinformatician, but it is rather difficult for the personnel expected to be at a clinical setting. The proposed setup would require not only a sequencing machine (Illumina), but also access to a HPC and a trained bioinformatician or computer-scientist to run the actual analysis. A better bridge could be attained by adding a GUI to the pipeline, and make sure it can run on a high-end laptop or equivalent to minimize the gap between bioinformatics and the clinical health microbiology health labs. Both CARD and ResFinder have added GUIs and web-services to make that bridge.

The pipeline proposed is composed of AMRFinderPlus with some postprocessing logic to infer phenotypes. The pipeline has been ISO-accredited, which adds some value to the workflow. However, it seems that the de novo assembly is not part of the ISO-accredited workflow, even though the assembly itself explains a lot of the discrepancies.

Similar pipelines have been developed in the field of bioinformatics, of which there should at least be a comparison towards CARD and ResFinder. Additionally, both CARD and ResFinder are able to align raw reads directly, and could therefore be used without the de novo assembly.

From ResFinder version 4 genotype-phenotype predictions have been added, which would make comparisons easier as the two pipelines have the same product (geno- and phenotype predictions). Thus, it would have been nice if the authors had evaluated their tool in comparison to other tools, as for example done here: <https://pubmed.ncbi.nlm.nih.gov/35662642/>.

The tool was evaluated only on Salmonella, where also other studies have shown excellent performance and it would naturally have been nice to see the performance for other species. Data set for this could be available from other studies where this have been done.

Reviewer #2 (Remarks to the Author):

In this paper the authors present a relatively simple, although useful, wrapper program for an established AMR gene prediction tool (AMRFinderPlus).

This wrapper, AbritAMR, is designed to support streamlining of AMR gene identification and reporting from bacterial genomes in clinical/public health lab settings.

What sets this paper apart from prior/similar bioinformatics papers is the considerable effort put towards validation, accreditation, and implementation of this tool.

This paper (pending some very minor fixes and inclusion of more details on implementation) has the potential to serve as a gold standard for how this sort of bioinformatics work should be performed given the growing use of pathogenomics in clinical/public health.

Minor Comments

- Introduction: it would be good to cite a paper or two at the start of paragraph 3 to show examples of how lack of standardisation means AMR genomics can have highly varied results from different labs (e.g., <https://www.ncbi.nlm.nih.gov/pmc/articles/PMC8549354/>)

- Methods: In terms of the previously accredited/validated genomics workflow: are the QC requirements values based on a specific analysis (or more reasonable heuristics)? I'm also assuming this genomics workflow also has some form of negative/positive controls included too, if so, can the details of these be mentioned? Acknowledging it isn't the main focus on this manuscript but it would be a great resource if this workflow was deposited into an open protocol repository such as protocols.io

- Methods: regarding the enhanced subclass information you've generated are you planning on continually updating this? How often do you think it would need updated?

- Results: how did the false negatives breakdown in terms of plasmid-borne vs chromosomal AMR genes?

- Results: report creation, given the increased interest in evidence-based report creation (e.g., elixir, Crisan et. al., TB reports) it would be great if the author's could include a bit more detail about the implementation process and consequent report modifications performed. Were requested changes consistent across different staff roles?

- Discussion: discussion of discrepancies should also refer to the general challenges of short-read methods for analysis of plasmid-borne AMR

- Figure 2: resolution seems a bit low in review PDF

- Figure 3: might be clearer/simpler if the "de novo assembly -> detect AMR genes -> output" boxes were shared between A and B. Makes it clearer that its the same process after read generation (by sequencing or simulation). It would also remove redundancy in the figure panels.

- Codebase: The repository installation instructions could do with being updated as it doesn't mention the conda package for arbitAMR that the developers have created. The current package name for the AMRFinderPlus dependency also needs updated to `ncbi-amrfinderplus` instead of `ncbi-amrfinder`. Would be nice to include a little documentation on running the tests to ensure users have correctly installed the tool.

- Codebase: Bioconda version tag is ahead of the repository tags making it a bit unclear which specific version is being referred to.

- Codebase: Currently `abritamr`, `abritamr run` and `abritamr mdu` all error if invoked without any params, ideally this should print the help and exit.

- Codebase: Error messages for `abritamr mdu` wasn't very helpful when incorrect/too few params were supplied: "Something has gone wrong with your inputs. Please try again!". I know this is primarily an internal feature but it would be nice if this was documented a bit more (e.g., mdu_qc) as it would be a great feature for other folks in clinical/public health labs. This doesn't seem too unreasonable given that it is one of the features discussed in the paper.

- Codebase: This may be more a "me problem" rather than a code issue but (for sake of completeness) I did find batch mode errored on some systems when not using bash (e.g., zsh). For some reason, the subprocess/parallel command seems to escape the conda env on non-bash shells and error due to not finding `amrfinder`. However, this is likely just a configuration issue on my end.

Reviewer #3 (Remarks to the Author):

Congratulations to the authors on this body of work! The manuscript is very well written with logical flow.

abritAMR builds upon AMRFinderPlus to detect AMR determinants and classify them into antibiotic classes. While there exists other software tools that perform a similar task, abritAMR has an enhanced drug class classification scheme and outputs reports that are suitable for clinical and public health microbiology interpretation. Lastly, abritAMR infers phenotype from genotype for *Salmonella* spp. The authors also validated abritAMR by comparing the results to PCR testing and synthetic sequencing reads.

The methodology used is sound and does use two different methods of validation of abritAMR. The study is reproducible as data used are publicly available and other wet-lab methods are detailed in the text.

The figures and tables are generally very clear and help with the understanding of the text, especially Figure 3.

Major comments

I think the introduction paragraph would benefit from:

1. a paragraph that highlights the importance of ISO accreditation, particularly since it is in the title but not really discussed anywhere else. As someone naive to the ISO standards, it would be helpful to know what receiving accreditation means especially since ISO15189 is not free for viewing.
2. Review of existing and similar bioinformatics tools (e.g., ResFinder, CARD/RGI) and highlight why abritAMR is different.

Figure 4/5 - It is hard to tell the difference between the false negative (pink) and false positive (orange) colours when there are small bars, especially in Figure 4 C2 and Figure 5B. Consider using different warm toned colours that are more distinct?

Figure 4 - "expanded out in Panel D" - I don't see a Panel D?

General questions

Table S1 - Why were some not included in validation panel?

QC - do only the reads have QC parameters? What about the assemblies?

"The abritAMR bioinformatic platform takes a genome assembly from short-read data" - could it also accept long read / hybrid assemblies or are there plans to incorporate this in the future?

Minor comments

In the text of the paper, the software is named "abritAMR" (lowercase t), but in the GitHub the software is named "abriTAMR" (uppercase T)

Figure S3 to S10 - Unsure how the bars are being organised but perhaps organise the bars based on number of isolates / genes (Highest to lowest)?

Figure S4 - Consider changing the gene combinations into an upset plot so that it is easier to understand?

Figure S2 - This figure makes it very clear what the reporting logic is. Is the logic for trimethoprim-sulfamethoxazole supposed to be ==Trimethoprim AND ==Sulfonamide (instead of or)?

Is there a reason that some antibiotics are highlighted in red?

In the discussion, perhaps add a part about the potential effects on performance when using

different assembly methods?

RESPONSE TO REVIEWERS

NCOMMS-22-21749: "Bridging the gap between bioinformatics and the clinical and public health microbiology laboratory: An ISO-accredited genomics workflow for antimicrobial resistance"

New title (to suit journal requirements): **Establishing ISO-accredited genomics workflows for antimicrobial resistance**

REVIEWER 1	
This is a very well-written, but also quite long manuscript. I am personally a bit uncertain regarding the scientific novelty of the work. On one hand it is certainly admirable that the authors try to promote and write about bioinformatics pipelines for routine diagnostics. This requires a lot of work and it is often very difficult to get acknowledged in the scientific literature compared to for example descriptions of yet another outbreak. However, implementing WGS based ISO-accreditation is not novel. This has been done in several laboratories globally and has been running in some places for 5-6 years (Genome Trakr, Genomics England, etc.). On that account, I also miss that the authors at least acknowledge the ISO-standards within this area (https://www.iso.org/obp/ui/#iso:std:iso:23418:ed-1:v1:en).	We thank the reviewer for taking time to review this manuscript. From the point-of-view of a working public health microbiology laboratory, genomics for detection of AMR determinants has been one of the most difficult areas for implementation, validation and accreditation, and here we aim to share our learnings for other laboratories in these settings. We believe this manuscript may also be of interest to bioinformaticians developing such tools, and international organisations promoting standardisation and harmonisation of genomic and bioinformatic methods. We agree that ISO-accreditation of WGS is not novel in itself, although we believe that ISO-accreditation for AMR is far less common compared to accreditation for sequencing and typing methods. As suggested by reviewers 2 and 3, we believe the novelty of this work lies in the extensive and rigorous validation presented, including provision of a validation dataset for public use, and the adaptation of the AMRFinderPlus tool specifically for implementation in clinical and public health microbiology reporting, where no publicly-available tools are currently fit-for-purpose to use unmodified in such reports. We have further elaborated on the relevant ISO standards in the introduction (lines 65-70) and added the appropriate references.
Regarding the tool itself, I do believe the authors are overselling it a bit. As far as I can judge, this works on assemblies only and require command-line expertise. The proposed pipeline is easy	This tool is designed for laboratories with bioinformatics expertise (or at least scientists familiar with command-line tools), most likely high-throughput sequencing laboratories, where automation is required to minimise the manual handling and

to install and use for a bioinformatician, but it is rather difficult for the personnel expected to be at a clinical setting. The proposed setup would require not only a sequencing machine (Illumina), but also access to a HPC and a trained bioinformatician or computer-scientist to run the actual analysis. A better bridge could be attained by adding a GUI to the pipeline, and make sure it can run on a high-end laptop or equivalent to minimize the gap between bioinformatics and the clinical health microbiology health labs. Both CARD and ResFinder have added GUIs and web-services to make that bridge.	interpretation of AMR results. To clarify this, we have modified the introduction (lines 72-76) and the discussion (lines 272-278). Assembled genomes are the required input for AMRFinderPlus, hence this has been adopted for abritAMR. We do agree that accessible, web-based tools with GUIs are an essential requirement for smaller laboratories without dedicated bioinformatics expertise to implement genomics, and that CARD and ResFinder are certainly starting to fill that need. This tool is not intending to fill that gap, but we hope that the principles described in this paper of tailoring outputs for reporting in clinical & public health settings, and validation methods to gain accreditation, may also be applicable to these tools in the future. We have added a comment in the discussion to this effect (lines 272-278).
The pipeline proposed is composed of AMRFinderPlus with some postprocessing logic to infer phenotypes. The pipeline has been ISO-accredited, which adds some value to the workflow. However, it seems that the de novo assembly is not part of the ISO-accredited workflow, even though the assembly itself explains a lot of the discrepancies.	Given the target audience for this pipeline, and the AMRFinderPlus requirement for assembled genomes, we have elected not to include genome assembly in this pipeline. We do, however, agree that a GUI-based pipeline aimed at laboratories without significant bioinformatic expertise, should include an assembly step, but this is not the aim of the current version in this manuscript. We have validated the results with an alternative assembler (SKESA) that we commonly use for additional analyses, and found minimal differences (Supplementary Table 1).
Similar pipelines have been developed in the field of bioinformatics, of which there should at least be a comparison towards CARD and ResFinder. Additionally, both CARD and ResFinder are able to align raw reads directly, and could therefore be used without the de novo assembly.	As described above, that the novel parts of the work we would like to emphasise are the classification & reporting logic tailored for clinical & public health microbiology, plus the rigorous validation methods, including a validation dataset that could be used by other labs. As noted in the Discussion (lines 239-241), this approach would certainly have worked if the pipeline was based on other AMR databases and tools, such as CARD or ResFinder. Indeed, the ability to use raw reads as inputs may be advantageous in some settings, but was not a factor in our choice due to our existing automated sequence data processing (which includes genome assembly).

	We have added a comment in the discussion (lines 267-272) about the features of ResFinder and CARD-RGI for CPHM reporting, with limitations being similar to AMRFinderPlus for this specific setting. We have also discussed future directions of pipeline development in these settings, including labs without bioinformatic expertise and HPC facilities.
From ResFinder version 4 genotype-phenotype predictions have been added, which would make comparisons easier as the two pipelines have the same product (geno- and phenotype predictions). Thus, it would have been nice if the authors had evaluated their tool in comparison to other tools, as for example done here: https://pubmed.ncbi.nlm.nih.gov/35662642/.	We note that these features of ResFinder were not available when we commenced developing this pipeline, and agree that the development of such features is of great significance in advancement of AMR genomics. Given that the primary focus of this manuscript is the validation and accreditation of the pipeline, with genotype-phenotype predictions being a minor component of the current pipeline, plus the already substantial length of the manuscript (as the reviewer has pointed out), we have elected not to perform these comparisons at the current time. We aim to further expand the geno-pheno predictions and validate across other species, at which time we will compare to other tools, as suggested.
The tool was evaluated only on Salmonella, where also other studies have shown excellent performance and it would naturally have been nice to see the performance for other species. Data set for this could be available from other studies where this have been done.	Evaluation of genotype-phenotype correlation is currently underway for other species, hence not included in the current manuscript.
REVIEWER 2	
In this paper the authors present a relatively simple, although useful, wrapper program for an established AMR gene prediction tool (AMRFinderPlus). This wrapper, AbritAMR, is designed to support streamlining of AMR gene identification and reporting from bacterial genomes in clinical/public health lab settings. What sets this paper apart from prior/similar bioinformatics papers is the considerable effort put towards validation, accreditation, and implementation of this tool. This paper (pending some very minor fixes and inclusion of more details on implementation) has the potential to serve as a gold	We thank the reviewer for their positive comments, and identifying the key issues that this paper is intending to address.

standard for how this sort of bioinformatics work should be performed given the growing use of pathogenomics in clinical/public health.	
Introduction: it would be good to cite a paper or two at the start of paragraph 3 to show examples of how lack of standardisation means AMR genomics can have highly varied results from different labs (e.g., https://www.ncbi.nlm.nih.gov/pmc/articles/PMC8549354/)	Thank you for the suggestion, two additional references have been added demonstrating the difficulties comparing AMR results between laboratories (line 52).
Methods: In terms of the previously accredited/validated genomics workflow: are the QC requirements values based on a specific analysis (or more reasonable heuristics)? I'm also assuming this genomics workflow also has some form of negative/positive controls included too, if so, can the details of these be mentioned? Acknowledging it isn't the main focus on this manuscript but it would be a great resource if this workflow was deposited into an open protocol repository such as protocols.io	We have provided a detailed summary of the accredited sequencing workflow in the Supplementary Methods, noting the difficult balance between providing adequate information vs providing methods very specific to our local equipment and protocols. The Supplementary Methods includes details of controls (positive/negative/PhiX) and QC metrics used for assessment of sequencing runs and individual sequences.
Methods: regarding the enhanced subclass information you've generated are you planning on continually updating this? How often do you think it would need updated?	We aim to update the enhanced subclass database after each AMRFinderPlus database update. Ideally, this would become obsolete if the tools and databases such as AMRFinderPlus adopted suggestions for clinically-relevant reporting frameworks and classifications, but until this time, we propose to keep updating this database.
Results: how did the false negatives breakdown in terms of plasmid-borne vs chromosomal AMR genes?	For the comparison to PCR results, the three false negatives were for CMY (1 isolate), CTX-M (1 isolate), and CTX-M + CMY (1 isolate). These are all most commonly plasmid-borne, leaving open the possibility that these discrepancies due to plasmid dropout in culture, despite PCR and WGS being performed on different colonies from the same pure subculture plate (note added in discussion, lines 306-307).
Results: report creation, given the increased interest in evidence-based report creation (e.g., elixir, Crisan et. al., TB reports) it would be great if the authors could include a bit more detail about the implementation process and consequent report modifications performed. Were requested changes consistent across different staff roles?	More detail added on the staff and external stakeholders consulted on the report creation (lines 200-203). As most of the changes were in the bioinformatic and internal processes (i.e. changing how genes were classified to prevent need for manual classification by reporting scientists), we felt that the changes in reporting formats were not significant enough to focus on in this particular manuscript.

Discussion: discussion of discrepancies should also refer to the general challenges of short-read methods for analysis of plasmid-borne AMR	We wholeheartedly agree with the reviewer that short-read sequencing is indeed a significant limitation for detection of transmission of plasmid-borne AMR genes, and one of the major hurdles in pathogen genomics to be overcome in the next few years. However, the focus of this paper on detection of AMR mechanisms regardless of their location (chromosomal or plasmid/MGEs), which we believe is not significantly impacted by short- vs long-read sequencing. As such, we have elected not to mention these challenges in our discussion.
Figure 2: resolution seems a bit low in review PDF	Thank you for bringing to our attention, this has been rectified.
Figure 3: might be clearer/simpler if the "de novo assembly -> detect AMR genes -> output" boxes were shared between A and B. Makes it clearer that its the same process after read generation (by sequencing or simulation). It would also remove redundancy in the figure panels.	Thank you for the suggestion – we have made some modifications to Figure 3 to declutter the figure.
Codebase: The repository installation instructions could do with being updated as it doesn't mention the conda package for arbitAMR that the developers have created. The current package name for the AMRFinderPlus dependency also needs updated to `ncbi-amrfinderplus` instead of `ncbi-amrfinder`. Would be nice to include a little documentation on running the tests to ensure users have correctly installed the tool.	Thank you, these have been updated.
Codebase: Bioconda version tag is ahead of the repository tags making it a bit unclear which specific version is being referred to.	Completed.
Codebase: Currently `abritamr`, `abritamr run` and `abritamr mdu` all error if invoked without any params, ideally this should print the help and exit.	Thank you for the suggestion, this has been implemented.
Codebase: Error messages for `abritamr mdu` wasn't very helpful when incorrect/too few params were supplied: "Something has gone wrong with your inputs. Please try again!". I know this is primarily an internal feature but it would be nice if this was documented a bit more (e.g., mdu_qc) as it would be a great feature for other folks in clinical/public health labs. This doesn't	Thank you for the suggestion, this has been implemented.

seem too unreasonable given that it is one of the features discussed in the paper.	
Codebase: This may be more a "me problem" rather than a code issue but (for sake of completeness) I did find batch mode errored on some systems when not using bash (e.g., zsh). For some reason, the subprocess/parallel command seems to escape the conda env on non-bash shells and error due to not finding `amrfinder`. However, this is likely just a configuration issue on my end.	We attempted to reproduce the error, but were unable to do so. If you have ongoing problems, we would be happy to investigate further if you are able to provide more details (e.g. file an issue on GitHub). Suggest you may also want to try to Docker (kristyhoran/abritAMR) alternative rather than conda if that suits you.
REVIEWER 3	
Congratulations to the authors on this body of work! The manuscript is very well written with logical flow. abritAMR builds upon AMRFinderPlus to detect AMR determinants and classify them into antibiotic classes. While there exists other software tools that perform a similar task, abritAMR has an enhanced drug class classification scheme and outputs reports that are suitable for clinical and public health microbiology interpretation. Lastly, abritAMR infers phenotype from genotype for Salmonella spp. The authors also validated abritAMR by comparing the results to PCR testing and synthetic sequencing reads. The methodology used is sound and does use two different methods of validation of abritAMR. The study is reproducible as data used are publicly available and other wet-lab methods are detailed in the text. The figures and tables are generally very clear and help with the understanding of the text, especially Figure 3.	We sincerely thank the reviewer for their very positive comments.
I think the introduction paragraph would benefit from: 1. a paragraph that highlights the importance of ISO accreditation, particularly since it is in the title but not really	1. Thank you for the suggestion, we have elaborated on ISO-accreditation processes and added a reference from the ISO website that explains what the ISO standards mean for medical laboratories (lines 65-70).

discussed anywhere else. As someone naive to the ISO standards, it would be helpful to know what receiving accreditation means especially since ISO15189 is not free for viewing. 2. Review of existing and similar bioinformatics tools (e.g., ResFinder, CARD/RGI) and highlight why abritAMR is different.	2. We have given details of the limitations of the ResFinder and CARD-RGI tools, similar to AMRFinderPlus, in the discussion (lines 267-272). Essentially, both tools have the same limitations as AMRFinderPlus from the CPHM point-of-view, and the abritAMR pipeline could have been developed using either tool as an alternative.
Figure 4/5 - It is hard to tell the difference between the false negative (pink) and false positive (orange) colours when there are small bars, especially in Figure 4 C2 and Figure 5B. Consider using different warm toned colours that are more distinct?	Figures 4 and 5 have been updated to show greater differentiation between false negatives and false positives.
Figure 4 - “expanded out in Panel D” - I don’t see a Panel D?	Thank you for identifying this discrepancy, we have amended the Figure 4 legend (changed ‘Panel D’ to ‘Panel C2’).
Table S1 - Why were some not included in validation panel?	Some AMR genes were uncommon amongst RefSeq genomes, and we were unable to identify a suitable reference genome (that also fit the most common species sequenced in our setting) that included these groups. As such, a small number of the AMR gene groups were not included in the validation panel. We have added a comment at the end of Supplementary Table 1 to explain this.
QC - do only the reads have QC parameters? What about the assemblies?	Quality assessment of genome assemblies (e.g. number of contigs or N50) is not usually required, as sequencing validation and accreditation was developed to ensure optimal assemblies for the purpose of gene detection; the key quality parameter for genome assembly is whether the assembly yields a complete MLST profile (where a scheme is available); incomplete profiles are flagged for further manual review. As with other high-throughput sequencing labs, our automated sequence QC and processing pipeline includes genome assembly using a single assembler for most routine workflows (in our case, SPAdes). These details have been expanded in the Supplementary Methods.
“The abritAMR bioinformatic platform takes a genome assembly from short-read data” - could it also accept long read / hybrid assemblies or are there plans to incorporate this in the future?	Clarified – abritAMR can take fasta files from short-read data, long-read data or hybrid assemblies (lines 350-351).

In the text of the paper, the software is named “abritAMR” (lowercase t), but in the GitHub the software is named “abriTAMR” (uppercase T)	This has been fixed, thanks for bringing to our attention.
Figure S3 to S10 - Unsure how the bars are being organised but perhaps organise the bars based on number of isolates / genes (Highest to lowest)?	This has been modified as requested.
Figure S4 - Consider changing the gene combinations into an upset plot so that it is easier to understand?	We thank the reviewer for the suggestion. We have attempted to wrangle these data into an upset plot, but found that it did not make the results any clearer, as there were too many variable combinations to make the upset plot interpretable.
Figure S2 - This figure makes it very clear what the reporting logic is. Is the logic for trimethoprim-sulfamethoxazole supposed to be ==Trimethoprim AND ==Sulfonamide (instead of or)? Is there a reason that some antibiotics are highlighted in red?	Thank you to the reviewer for their detailed examination of these figures! The presence of resistance genes to either of trimethoprim or sulfamethoxazole renders the organism resistant to the combination drug in almost all cases, hence this is meant to be ‘or’. Red text was from a previous figure version and has been removed, with thanks.
In the discussion, perhaps add a part about the potential effects on performance when using different assembly methods?	This has been added to the Discussion (lines 307-310)

EDITORIAL REQUIREMENTS

Editorial policy checklist	Complete
Reporting summary	Complete
Code and software submission checklist	Complete Note that code is publicly available (reviewers have already accessed), and the accession numbers for the dataset used are included in the manuscript
Data availability statement with accessions and hyperlinks	Included in manuscript
Code availability statement	Included in manuscript
Source data	Not applicable (all data points shown in bar graphs); cannot be replaced with box and whisker plots as categorical data
ORCID	Included for Corresponding Author

Reviewer #1 (Remarks to the Author):

As far as I can judge the authors have nicely addressed all comments.

Reviewer #2 (Remarks to the Author):

The authors have fully addressed my review and have produced a useful and important piece of work.